# Combinatorial summation of Feynman diagrams

Evgeny Kozik [1] ✉

Feynman's diagrammatic series is a common language for a formally exact theoretical description of systems of infinitely-many interacting quantum particles, as well as a foundation for precision computational techniques. Here we introduce a universal framework for efficient summation of connected or skeleton Feynman diagrams for generic quantum many-body systems. It is based on an explicit combinatorial construction of the sum of the integrands by dynamic programming, at a computational cost that can be made only exponential in the diagram order on a classical computer and potentially polynomial on a quantum computer. We illustrate the technique by an unbiased diagrammatic Monte Carlo calculation of the equation of state of the $2D$ $SU(N)$ Hubbard model in an experimentally relevant regime, which has remained challenging for state-of-the-art numerical methods.

From quantum electro- and chromodynamics to nuclear and condensed matter physics, Feynman's diagrammatic technique is arguably the most universal approach to describing quantum many-body systems. Scattering amplitudes and decay rates, masses of elementary particles, thermodynamic observables or correlation functions are expressed formally exactly as an infinite sum of all connected Feynman diagrams of the many-body perturbation theory[1]. Each diagram represents a formula for computing one term in this sum, which in the simplest case consists of a product of one-particle non-interacting Green's functions $G^0$ and a number $n$ (the diagram order) of interaction vertices $V$ (For clarity, we confine ourselves to the case of pairwise interactions; see Fig. 1a,b). Integrated over all internal variables. The accuracy of computing a physical observable in a diagrammatic calculation relies on one's ability to evaluate all terms of the series to a sufficiently high order $n$. This is accomplished by the diagrammatic Monte Carlo (DiagMC) approach[2–5] through sampling the series stochastically, at the expense of a known and controllable statistical uncertainty. The key advantage of this approach is that the diagrams are defined directly in the thermodynamic limit, circumventing the need to extrapolate the result with the system size, which is typically hard in unbiased quantum Monte Carlo methods due to the negative sign problem[6,7]. However, the accuracy of evaluating a high-order contribution is inhibited by a factorially increasing with $n$ number of diagrams and hence exploding Monte Carlo variance. Indeed, in correlated regimes all ~ $n!$ order-$n$ diagrams are typically of comparable magnitude[8], which largely negates the chief advantage of Monte Carlo

– that of importance sampling. This suggests that the summation over diagram topologies and indices on which there is only weak dependence could be done deterministically to a similar effect, or more efficiently if such summation could be performed faster than in $\mathcal{O}(n!)$ steps. For fermionic systems, where the diagrams have alternating signs, this also helps lower the Monte Carlo variance[9–12]. Crucially, if the computational cost could be reduced to exponential in $n$, it was shown in ref. 11 (with an extension to divergent series [13], if necessary) that the computational time would scale only polynomially with the inverse of the desired error bound.

An instructive example is the $SU(N)$-symmetric Hubbard model for $N$ species of fermions. An approximate large-$N$ (pseudo-)spin symmetry emerges in multi-orbital condensed matter systems due to orbital degeneracy[14,15]. It is relevant to the description of, e.g., transition-metal oxides and orbital-selective Mott and superconducting transitions[14,16–19], graphene and twisted bilayer graphene[15,19–21], and is expected to harbour exotic phases of matter, such as topologically non-trivial spin liquids[22]. However, it poses a serious challenge for precision numerical methods owing to the additional exponential scaling of the Hilbert space with $N$, aggravating the sign problem[23]. Existing DiagMC algorithms based on determinantal summation of connected diagrams[9,10], which are very efficient in the $SU(2)$ case, are limited by the rigid structure of the determinant: the ~ $N^2/2$ choices for each of the $n$ interaction lines increase the computational cost of summing all diagrams of order $n$ by a factor ~ $(N^2/2)^n$. The recent studies by Ibarra-García-Padilla et al.[24] using the

---

[1]Department of Physics, King's College London, London, UK. ✉e-mail: evgeny.kozik@kcl.ac.uk

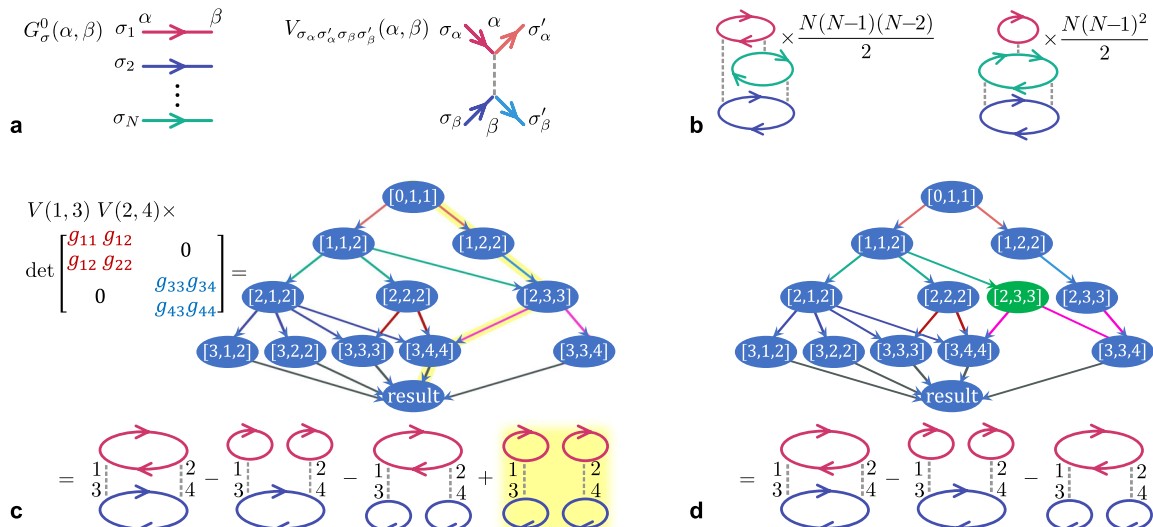

**Fig. 1 | Illustration of the algorithm. a** Graphical elements of the diagrammatic theory: the non-interacting Green's function (propagator) $G_\sigma^0(\alpha, \beta)$ and interaction $V_{\sigma_\alpha \sigma'_\alpha \sigma_\beta \sigma'_\beta}(\alpha, \beta)$, where $\sigma = 1, \ldots N$ are the spin indices and the vertex index $\alpha \equiv (i_\alpha, \tau_\alpha)$ is the combined lattice coordinate $i_\alpha$ and imaginary time $\tau_\alpha$. **b** Example of connected diagrams with their spin-/topology-dependent multiplicities $M$. **c** All closed-loop diagrams at order $n = 2$ for $N = 2$, generated by the determinant of the block-diagonal in this case matrix $g_{\alpha\beta} = G^0(\alpha, \beta)$ and computed following Eq. (2) by the directed graph. The nodes are labelled by $[l, h, e]$, where $l$ is the level of the graph, $h$ the head (smallest element) in the current cycle, and $e$ the current element within the cycle. Node [0, 1, 1] has the value 1. An edge going from $[l, h_1, e_1]$ to $[l + 1, h_2, e_2]$ multiplies the value of node $[l, h_1, e_1]$ by $g_{e_1 e_2}$ if $h_1 = h_2$ (continuing the cycle), or by $-g_{e_1 h_1}$ if $h_2 = e_2$ (closing the cycle), and adds the result to node $[l + 1, h_2, e_2]$. The "result" node stands for $[2n, 2n + 1, 2n + 1]$. Each path along the edges generates a single diagram, and there is a factorial number of them, but the number of edges, and thus the cost of computing the sum, is only $\mathcal{O}(n^4)$. The highlighted path generates the unwanted disconnected diagram. **d** The disconnected diagram is eliminated by cutting the corresponding path in the graph, which amounts to duplicating node [2, 3, 3] and re-routing some of its edges through the copy.

determinantal quantum Monte Carlo (DQMC)[25,26] and numerical linked-cluster expansion (NLCE)[27,28] methods at finite temperature, and by Feng et al.[29] using the auxiliary-field quantum Monte Carlo (AFQMC) method[30] with improvable constraints[31] at zero temperature, revealed a rich phase diagram of the $SU(N)$ Hubbard model at $N = 3$ and density $\langle n \rangle = 1$. At large $N$, however, unbiased numerical methods are currently outperformed by experimental realisations of the system with ultra-cold alkaline-earth-like atoms in optical lattices[32–38]—analogue quantum simulators[39,40]—in accessing the regimes of low temperatures and strong correlations[37,38].

Here we develop a framework for efficient evaluation of Feynman's diagrammatic series of arbitrary structure by deterministic summation of all diagram integrands. The approach is based on an explicit combinatorial construction of each term in the sum, one Green's function at a time, whereby at each step the result is maximally factorised into sums of single Green's functions by dynamic programming. Specifically, the result takes the form of a directed graph (Fig. 1d), with each node being a sum of contributions from its incoming edges, and each edge conveying the value of the previous node multiplied by a Green's function. In this approach, the $SU(N)$ symmetry is accounted for by merely an additional multiplication of certain edges by a constant factor, while all connected diagrams of order $n$ can be summed in at most $\mathcal{O}(n^3 4^n)$ steps independently of $N$. This is reduced for the special case of $N = 1$ (spinless fermions) and $SU(2)$ Hubbard model to $\mathcal{O}(n^3 3^n)$. The factorisation of the sum, which serves to minimise the number of repeated uses of each Green's function value, is the essence of the speed-up. As a byproduct, the result is also symmetrised over $n!$ permutations of the interaction lines and $2^n$ inversions of their endpoints, helping to further reduce the variance of the DiagMC evaluation of the series. However, the connected topologies with ordered interaction ends can be summed in $\mathcal{O}(n^3 3^n)$ steps in the $SU(N)$ case, and those with completely ordered vertices in only $\mathcal{O}(n^2 2^n)$ operations. Following ref. 11 (and[13] in the case of a divergent series), the exponential computational cost of this

approach implies polynomial scaling of the calculation time with the required accuracy. The approach admits a vector formulation, which is potentially suitable for a realisation on a quantum computer with a further exponential speed-up.

We apply the combinatorial summation (CoS) technique to a calculation of the equation of state (EoS) of the $2D$ $SU(N)$ Hubbard model in the case of $N = 6$, which is relevant for experiments, but hard for numerical methods. We first address the low-temperature regime studied very recently by Pasqualetti et al.[38], where the system was realized using the 6 nuclear spin states of $^{173}$Yb atoms loaded in an optical lattice, and the experimentally obtained EoS was cross-benchmarked against unbiased DQMC calculations. The range of the CoS technique is then explored by extending the calculations to lower temperatures and greater interaction strengths, where the sign problem is known to rapidly intensify[23], and experimental data for $N = 6$ cannot be reliably captured by numerical methods[37]. At the low-temperature/strong-coupling boundary of the studied regime, traits of a developing (pseudo-)gapped state are observed.

## Results

For simplicity, let us confine ourselves to the fermionic $SU(N)$ Hubbard model from the start, which is defined by the Hamiltonian

$$\hat{H} = -t \sum_{\langle i,j \rangle, \sigma} \left( \hat{c}_{i\sigma}^\dagger \hat{c}_{j\sigma} + H.c. \right) + \frac{U}{2} \sum_{i, \sigma_1 \neq \sigma_2} \hat{n}_{i\sigma_1} \hat{n}_{i\sigma_2} - \mu \sum_{i, \sigma} \hat{n}_{i\sigma}. \quad (1)$$

Here the operators $\hat{c}_{i\sigma}^\dagger$ and $\hat{c}_{i\sigma}$ create and annihilate a fermion on site $i$ with the spin $\sigma = 1, \ldots, N$, respectively, $\hat{n}_{i\sigma} = \hat{c}_{i\sigma}^\dagger \hat{c}_{i\sigma}$ is the number operator, $t$ the hopping amplitude, $U$ the on-site interaction, $\mu$ the chemical potential, and $\langle i, j \rangle$ implies that the summation is over nearest-neighbour lattice sites. A thermodynamic observable, such as, e.g., the average potential energy, is expressed diagrammatically (Fig. 1a, b) as the sum of all connected closed-loop diagrams obtained by linking vertices $\alpha$, representing a point on the lattice $i_\alpha$ and in

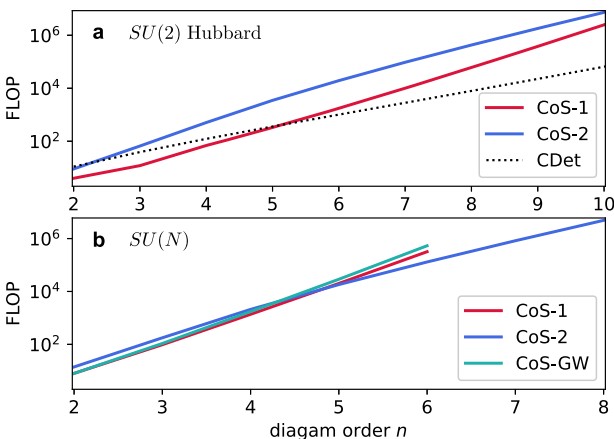

**Fig. 2 | Number of floating-point operations (FLOP) required to evaluate the sum of the integrands of all Feynman diagrams of order $n$. a** For connected diagrams of the $SU(2)$ Hubbard model summed by the algorithm of Section Approach 1: modification of the determinant (CoS-1) and that of Section Approach 2: constructing connected diagrams from the start (CoS-2), for which the curve closely follows $\approx n^3 3^n/8$; the reference dotted line corresponds to the state-of-the-art implementation of the CDet algorithm[10] with fast principal minor calculation[45] ($\sum_{l=0}^{n-2} 2^{l+1}(n-l-1)^2 + 3^n$); **b** for connected diagrams in the $SU(N)$ case, with the curve for CoS-2 following $\approx n^3 4^n/7$. Shown as CoS-GW is the computational cost of summing the skeleton (bold-line) series in terms of the renormalised Green's function $G$ and screened interaction $W$. Source data are provided as a Source Data file.

imaginary time $\tau_\alpha$, by the interaction lines $V_{\sigma_\alpha \sigma'_\alpha \sigma_\beta \sigma'_\beta}(\alpha, \beta) = \frac{U}{2} \delta_{\sigma_\alpha, \sigma'_\alpha} \delta_{\sigma_\beta, \sigma'_\beta} (1 - \delta_{\sigma_\alpha, \sigma_\beta}) \delta_{i_\alpha, i_\beta} \delta(\tau_\alpha - \tau_\beta)$ and non-interacting propagators (Green's functions) $G^0_\sigma(\alpha, \beta) = - \langle \mathcal{T} \hat{c}_{i_\beta \sigma}(\tau_\beta) \hat{c}^\dagger_{i_\alpha \sigma}(\tau_\alpha) \rangle_0$, where $\mathcal{T}$ is the time ordering operator and the statistical average $\langle \ldots \rangle_0$ is taken with the Hamiltonian at $U = 0$, and summing or integrating the result over all its $\sigma_\alpha, i_\alpha, \tau_\alpha$ variables. It is well known—and used in finite-size determinant diagrammatic Monte Calro methods[41,42]—that the sum of all combinations of $n$ interactions with the propagators is generated by the determinant of a $2n \times 2n$ matrix $g_{\alpha\beta} = G^0(\alpha, \beta)$, $\alpha, \beta = 1, \ldots, 2n$ (the spin indices are omitted for clarity), multiplied by the corresponding values of $V(\alpha, \beta)$. This way the (2n)! terms can be produced extremely efficiently in $\mathcal{O}(n^3)$ operations, but having to eliminate the unwanted disconnected diagrams from the determinant afterwards requires at least an exponential number of steps[10]. Our strategy, in contrast, will be to not generate the disconnected diagrams from the start.

## Combinatorial summation for the determinant

A good starting point is the algorithm for division-free calculation of the determinant[43] based on its permutation cycle decomposition. In terms of the cycle covers $\mathcal{C} = (\alpha_1, \alpha_2, \ldots \alpha_{c_1}) \ldots (\alpha_{c_{m-1}+1}, \alpha_{c_{m-1}+2}, \ldots, \alpha_{2n})$, representing an ordered sequence of matrix indices (called elements) grouped into $m$ cycles by the parenthesis, the determinant becomes

$$\det\{g_{\alpha\beta}\} = \sum_\mathcal{C} \text{sign}\,\mathcal{C} \cdot \text{weight}\,\mathcal{C}, \quad (2)$$

where $\text{sign}\,\mathcal{C} = (-1)^{2n+m}$ and $\text{weight}\,\mathcal{C} = (g_{\alpha_1 \alpha_2} \ldots g_{\alpha_{c_1} \alpha_1}) \ldots (g_{\alpha_{c_{m-1}+1} \alpha_{c_{m-1}+2}} \ldots g_{\alpha_{2n} \alpha_{c_{m-1}+1}})$. For instance, the cycle cover $\mathcal{C} = (1\,2\,5\,3)(4\,8\,7)(6)$ has $\text{sign}\,\mathcal{C} = (-1)^3$ and $\text{weight}\,\mathcal{C} = (g_{12} g_{25} g_{53} g_{31})(g_{48} g_{87} g_{74})(g_{66})$. In this form, one easily recognises Feynman's rules for constructing the diagrams[1], with the cycles corresponding to fermionic loops. It is useful to view building each $\mathcal{C}$, one element at a time, by an ordered walk of $2n$ steps, where at each step $l$ the current element is $e$ and the

new element $e'$ is selected according to some rules, while the current sign $\mathcal{C} \cdot$ weight $\mathcal{C}$ is multiplied by $g_{ee'}$, as well as by an additional $-1$ when the cycle is closed. An expression like Eq. (2) is then evaluated as a sum over all such walks. The central observation[43] is that, when different walks are executed in parallel, there will be many for which the step $l$ is identical. Thus, before step $l$ the weights of all such walks constructed up to this point can be combined, and the multiplication by $g_{ee'}$ applied to the sum. This suggests linking all walks in a graph, such as that in Fig. 1c, where the result of the summation before each step is stored in the nodes and the steps are the edges. An optimal structure of the graph minimises the number of times the multiplication by $g_{ee'}$ needs to be performed, and finding it is the task of dynamic programming. In the case of the determinant, the total number of edges can be made only polynomial in $n$.

A unique element $e$ must appear in $\mathcal{C}$ only once, which in general makes step $l$ dependent on all the steps before it. However, it was demonstrated in ref. 43 that all terms with repeated elements will cancel out due to the sign structure, provided the lowest element in each cycle within $\mathcal{C}$, called the cycle head $h$, is placed at the beginning of the cycle and is present in $\mathcal{C}$ only once. Then, for each $\mathcal{C}$ with a repeated element, there will be exactly one $\mathcal{C}'$ such that weight $\mathcal{C}' =$ weight $\mathcal{C}$, but the number of its cycles differs by one, i.e. sign $\mathcal{C}' = -$ sign $\mathcal{C}$. This is straightforward to ensure if, at each step $l$, the head of the current cycle $h$ is stored alongside the current element $e$, and the next step is either to any other element $e' > h$ within the cycle, or starts a new cycle with $h' > h$ and $e' = h'$, so that the cycle heads are picked in ascending order. Therefore, each unique node must carry the three numbers $[l, h, e]$, $l = 0, \ldots 2n$; $h, e = 1, \ldots 2n + 1$. The resulting graph, computing the determinant in $\mathcal{O}(n^4)$ floating-point operations[43], is illustrated in Fig. 1c.

## Approach 1: modification of the determinant

Since there is a one-to-one correspondence between a particular path in the graph and the diagram it generates, the task of omitting the disconnected diagrams from the determinant can be formulated as that of identifying the corresponding paths and eliminating them selectively. Preserving all other paths is in principle accomplished by duplicating certain nodes along the unwanted paths and re-routing the paths to be kept through the copies, as in the example in Fig. 1d. This suggests that the information $[l, h, e]$, which uniquely identifies the nodes in the determinant, is incomplete for a diagrammatic series obeying more general rules, and the node label must be extended by some additional record $\mathcal{R}$. If what constitutes $\mathcal{R}$ is identified, the right graph can be constructed from the start by the algorithm of the previous section with the modification that the two nodes $[l_1, h_1, e_1] \otimes \mathcal{R}_1$ and $[l_2, h_2, e_2] \otimes \mathcal{R}_2$ are considered one and the same if $\mathcal{R}_1 = \mathcal{R}_2$ in addition to $l_1 = l_2$, $h_1 = h_2$, $e_1 = e_2$. It is desirable that the information in $\mathcal{R}$ is kept minimal to prevent spawning redundant nodes, but a sub-optimal graph can always be pruned in the end, without changing its value, by merging all nodes with equal $[l, h, e]$ that connect to the same nodes at the next level.

A disconnected diagram is produced when not all of its cycles (fermionic loops) end up linked by the interaction lines. Thus, an obvious choice for $\mathcal{R}$ is the list of vertices visited until the current step and grouped together according to their cycles, with the groups merged at each step if the corresponding cycles become linked by an interaction. Denoting each group by {...} and listing the current unfinished group last, the highlighted path in Fig. 1c would become $[0, 1, 1] \otimes \{1\} \rightarrow [1, 2, 2] \otimes \{1\}\{2\} \rightarrow [2, 3, 3] \otimes \{2\}\{13\} \rightarrow [3, 4, 4] \otimes \{13\}\{24\} \rightarrow$ result, and it is now obvious that it produces a disconnected diagram because the two groups in $\mathcal{R} = \{13\}\{24\}$ cannot be linked. Note that, for this choice of $\mathcal{R}$, the cancellation between terms with repeated elements, relied on in the calculation of the determinant, is in general between a connected and disconnected term. Thus it is generally necessary to also prohibit sequences $\mathcal{C}$ with repeated

elements. The cancellation can still be usefully employed in certain cases, as explained below.

For the $SU(N)$ Hubbard interaction in the form (1), where the same-spin coupling is excluded, the sum over different combinations of spin indices implies that each diagram comes with the spin- and topology-dependent multiplicity factor $M = \sum_{\sigma_1,\dots,\sigma_m} \prod_{\text{interactions}} (1 - \delta_{\sigma_i,\sigma_j})/2$, where $m$ is the number of loops and each interaction in the product connects a loop with spin $\sigma_i$ to that with spin $\sigma_j$, as in the example in Fig. 1b. A strength of our approach is that an arbitrary factor can be accounted for by merely (i) grouping the diagrams with the same $M$ together and (ii) multiplying the value of each node at the penultimate level $l = 2n - 1$ by $M$. To this end, we also store in $\mathcal{R}$ a matrix of connections between the cycles and prune the final graph to minimise its size.

Despite the combinatorial structure of $\mathcal{R}$, this algorithm is already efficient at diagram orders typically accessible in calculations. Indeed, since the Monte Carlo variance of integration over the vertex positions and times scales exponentially with diagram order $n$[11], in correlated regimes, only contributions from $n \lesssim 10$ can typically be evaluated with $< 10\%$ statistical error required for precision reconstruction of observables[44]. Figure 2 shows the actual number of floating point operations required to sum all connected diagrams of order $n$, with this approach labelled there as CoS-1. For the $SU(2)$ Hubbard model, where each diagram has multiplicity $M = 1$ and an efficient algorithm exists—connected determinant diagrammatic Monte Carlo (CDet)[10] with the fast principal minor algorithm[45]—CoS-1 already exhibits competitive performance. In the $SU(N)$ case, the computational cost of CoS-1 is independent of $N$ (for $N > 4$) and appears exponential for orders $n \lesssim 6$, although it is expected to eventually rise combinatorially. Nonetheless, at large $n$, CoS-1 is superseded by an approach of exponential complexity described below.

## Approach 2: constructing connected diagrams from the start

There is much freedom in how the graph summing a particular series is designed, and the following general principles can aid its efficiency: (i) allowing unwanted or unphysical sequences $\mathcal{C}$ might be useful if they cancel in the final sum, and (ii) walks can traverse the cycle covers $\mathcal{C}$ and be grouped in arbitrary order, provided all the required sequences are generated in the end. Principle (i) was key for computing the determinant, but it has another use here: we can formally allow on-site interactions between same-spin fermions in the Hamiltonian (1) since the resulting diagrams cancel. Instead of the topology-dependent factor $M$, the diagrammatic rules[1] for fully spin-symmetric interactions prescribe that each fermionic loop is multiplied by the number of spins, implying a multiplication of each node that closes a cycle merely by $N$. Although having to construct diagrams that cancel is a hindrance at lower orders, the simpler diagrammatic rules allow for a more efficient scaling at $n \gtrsim 5$–$6$.

Our recipe for organising the walks that constitute the graph has so far been borrowed from the determinant, forcing us to keep track in $\mathcal{R}$ of how different cycles are connected. This is not necessary if we reorganise the walks to generate only connected diagrams from the start. Since for generic Hamiltonians we cannot rely on the cancellation of terms with repeated elements, we at least must keep track of the elements visited up to the current step, $l$, $\mathcal{R} = \{e_1, e_2, \dots e_l\}$, and ban adding $e$ to $\mathcal{C}$ if $e \in \mathcal{R}$. Demoting the role of $h$ in the node label $[l, h, e] \otimes \{e_1, e_2, \dots e_{l-1}, e\}$ to being merely the first element in the current cycle, we can generate only connected diagrams if each new cycle starts with an element that is paired by an interaction to one of the already visited ones $e_i \in \{e_1, e_2, \dots e_{l-1}, e\}$, e.g. the smallest in $\mathcal{R}$ that is not already paired, for uniqueness. It is easy to see that the number of floating point operations in this graph is only exponential, $\mathcal{O}(n^3)4^n$, and that the information about visited elements carried in $\mathcal{R}$ is minimal for this order of traversing the sequences $\mathcal{C}$, i.e. the graph cannot be pruned any further. Indeed, at a given level $l$ the number of nodes, i.e. unique labels

$[l, h, e] \otimes \mathcal{R}$, is a product of the number of values of $h$, $e$ and $\mathcal{R}$, given by $\mathcal{O}(l)$, $\mathcal{O}(2n - l)$ and $\mathcal{O}((2n)!/l!(2n - l)!)$ respectively. Each node is linked to $\mathcal{O}(n)$ nodes at the level $l + 1$, and thus the total number of edges is obtained by summing $n\, l(2n - l)(2n)!/l!(2n - l)!$ over all levels $l$, resulting in the exponential scaling. The practical computational cost of this algorithm, labelled CoS-2, is shown for the $SU(N)$ case in Fig. 2b.

In systems where there is no non-trivial factor associated with each fermionic loop, as, e.g., in the $SU(2)$ Hubbard model, or for $N = 1$, cancellations between cycle covers with repeated elements can still be utilised to reduce the cost further to $\mathcal{O}(n^3)3^n$. To this end, $\mathcal{R}$ only needs to store, for each interaction line, the number of its vertices—zero, one or two—that have already been visited. Since there is no record of which of the two vertices of an interaction has been visited, both options for the element that starts a new cycle need to be allowed, with the cycle cover that ends up repeating the vertex cancelling out, as in the case of the determinant. The complexity of this algorithm is plotted for the $SU(2)$ case in Fig. 2a.

Finally, sums of skeleton (bold-line) diagrams in arbitrary channels can be straightforwardly generated in our approach. For instance, the computational cost of producing an expansion in terms of the full (interacting) Green's function $G$ and screened interaction $W$[46] by a simple extension of the CoS-2 algorithm is plotted in Fig. 2b as CoS-GW. The challenge of restricting the series to irreducible diagrams in both channels is met here by supplementing the nodes in the CoS-2 graph with a record $\mathcal{R}$ that keeps track of connectivity—as in the CoS-1 approach of the previous section—when any two propagators in a cycle or two interaction lines between different cycles are cut. Curiously, there is no notable cost increase relative to the CoS-1 algorithm for connected diagrams. The versatility of the CoS platform could enable more efficient algorithms for skeleton series in the future.

## Vector variant and quantum speed-up

The CoS algorithm can be cast in a vector form, in which the graph remains of a polynomial in $n$ size with the nodes uniquely identified by $[l, h, e]$ (as in Fig. 1c), but operates on a vector of values $|\psi\rangle = \sum_{\mathcal{R}} v_{\mathcal{R}} |\mathcal{R}\rangle$, with the floating-point numbers $v_{\mathcal{R}}$ used to construct its value and the vectors $|\mathcal{R}\rangle$ of an orthonormal set $\{|\mathcal{R}\rangle\}$ being responsible for filtering valid diagram configurations. For the algorithm of Section Approach 2: constructing connected diagrams from the start (CoS-2), $|\mathcal{R}\rangle$ is a direct product of $2n$ orthonormal states $|0\rangle$ or $|1\rangle$, indicating whether an element $e$ has been visited ($|1\rangle_e$) or not ($|0\rangle_e$), so that $\mathcal{R} = \{e_1, e_2, \dots e_l\}$ corresponds to $|\mathcal{R}\rangle = |1\rangle_{e_1} |1\rangle_{e_2} \dots |1\rangle_{e_l} |0\rangle_{e_{l+1}} \dots |0\rangle_{e_{2n}}$. The subspace of $\{|\mathcal{R}\rangle\}$ to be passed on by each edge is selected using the projection operators $\hat{P}_e^0 = |0\rangle_e \langle 0|_e$, $\hat{P}_e^1 = |1\rangle_e \langle 1|_e$, $\hat{\bar{P}}_e^0 = |1\rangle_e \langle 0|_e$ and $\hat{\bar{P}}_e^1 = |0\rangle_e \langle 1|_e$. Specifically, an edge adding a new element within a cycle, $[l, h, e_1] \to [l + 1, h, e_2]$, must project out all contributions in which the element $e_2$ has already been visited before multiplying the result by $g_{e_1 e_2}$ and adding it to the next node,

$$
[l, h, e_1] \to [l + 1, h, e_2] : \\
|\psi_2\rangle := |\psi_2\rangle + g_{e_1 e_2} \hat{\bar{P}}_{e_2}^0 |\psi_1\rangle, \tag{3}
$$

where $|\psi_1\rangle$ and $|\psi_2\rangle$ are the vectors stored in the nodes $[l, h, e_1]$ and $[l + 1, h, e_2]$, respectively. Edges that start a new cycle with an element $h_2$ act on the subspace in which $h_2$ is paired by an interaction to the lowest unpaired visited vertex in $|\mathcal{R}\rangle$,

$$
[l, h_1, e_1] \to [l + 1, h_2, h_2] : \\
|\psi_2\rangle := |\psi_2\rangle - g_{e_1 h_1} \hat{\bar{P}}_{h_2}^0 \hat{P}_{h_2}^1 \prod_{e < h_2} \left[ \hat{P}_{\bar{e}}^1 \hat{P}_e^1 + \hat{P}_e^0 \right] |\psi_1\rangle, \tag{4}
$$

where $|\psi_1\rangle$ and $|\psi_2\rangle$ are the vectors stored in the nodes $[l, h_1, e_1]$ and $[l + 1, h_2, h_2]$, respectively, and $\bar{e}$ is the vertex paired to $e$ by an

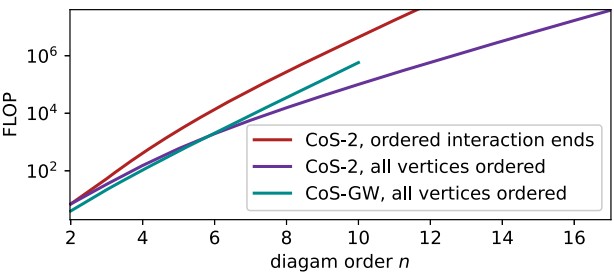

**Fig. 3 | Number of floating-point operations (FLOP) required to evaluate the sum of the integrands of all Feynman diagrams of order $n$ by the CoS-2 algorithm without inherent symmetrisation over vertex permutations: for connected diagrams in the $SU(N)$ case with ordered end points of interactions (CoS-2, ordered interaction ends) and fully ordered vertices (CoS-2, all vertices ordered).** Shown as CoS-GW, all vertices ordered is the computational cost of summing the nonsymmetrised skeleton (bold-line) series in terms of the renormalised Green's function $G$ and screened interaction $W$. Source data are provided as a Source Data file.

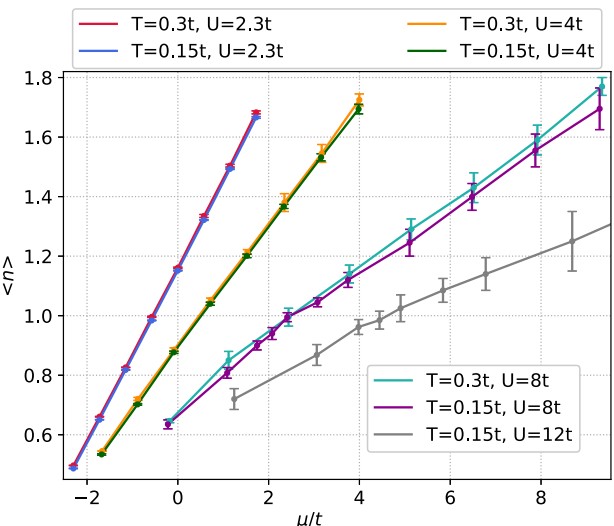

**Fig. 4 | Equation of state for the 2D $SU(N)$ Hubbard model with $N = 6$ for $T/t$ = 0.3, 0.15 and $U/t$ = 2.3, 4, 8, 12.** The error bars are a sum of the systematic error of resummation beyond the convergence radius and the propagated statistical (standard deviation) error of the series coefficients; see Methods. Source data are provided as a Source Data file.

interaction. Following this recipe, at the result node we obtain a pure state $|\psi_{\text{result}}\rangle = v|1\rangle_1|1\rangle_2 \dots |1\rangle_{2n}$ with $v$ being the value of the graph.

On a classical computer, the elementary base vectors have to be represented by two components, $|0\rangle = (1,1)^T/\sqrt{2}$, $|1\rangle = (1,-1)^T/\sqrt{2}$, implying that $|\psi\rangle$ is a $2^{2n}$-component vector, and the edges (3), (4) take $\mathcal{O}(4^n)$ floating-point operations to evaluate. Given that the number of edges scales as $\mathcal{O}(n^4)$, the computational cost of this approach, $\mathcal{O}(n^4 4^n)$, is a factor $\propto n$ higher than that of the CoS-2 algorithm. Nonetheless, an efficient processing of vector operations, e.g. by GPUs, could make the vector implementation faster in practice.

The ability to efficiently operate with vector superpositions makes the quantum computer a promising platform for this approach. To this end, the graph defines a quantum circuit processing the state $|\psi\rangle = \sum_{\mathcal{R}} |v_{\mathcal{R}}\rangle|\mathcal{R}\rangle$, where $|v_{\mathcal{R}}\rangle$ encodes the value and $|\mathcal{R}\rangle$ is represented by $2n$ qubits. Projections can be generally performed by unitary quantum gates, while the multiplication by the matrix elements of $g_{\alpha\beta}$ could be implemented, e.g., by quantum floating-point arithmetic[47,48].

Provided a practical quantum implementation incurs at most polynomial computational overheads, the $\mathcal{O}(n^4)$ graph could be evaluated in a polynomial number of operations on a quantum processor. The result could then be used in the Monte Carlo integration over vertex coordinates on a classical processor, similar to the quantum-classical approach[49]. An interesting possibility to explore is making the Metropolis sampling quantum as well[50,51], e.g., through a mapping of the graph value to the quantum eigenvalue/eigenvector problem[52], which could enable a further speed-up.

### Nonsymmetrised sum

Like the determinant, in its general form, the CoS approach sums not just all diagram topologies of a given order $n$ but their $2^n n!$ realisations with all permutations of the interaction lines and endpoints. An advantage of the explicit summation is that this symmetrisation can be suppressed if useful with a reduction of the computational cost. Specifically, since the Hubbard interaction is local in space and time, the symmetrisation over $2^n$ exchanges of the interaction end points does not affect the variance of the Monte Carlo sampling. In this case, ordering the arguments in each $V(\alpha, \beta)$ leads to a tangible speed-up of the DiagMC sampling. Indeed, by requiring that a node $[l,h,\bar{e}] \otimes \mathcal{R}$ is added to the graph only if $e$ (the vertex connected to $\bar{e}$ by an interaction) is already present in $\mathcal{R}$, the number of operations in the CoS-2 approach drops from $\mathcal{O}(n^3 4^n)$ to $\mathcal{O}(n^3 3^n)$. The actual computational cost shown in Fig. 3 (CoS-2, ordered interaction ends) follows this scaling.

Similarly, the summation over $n!$ permutations of the $n$ interaction lines is suppressed if a node $[l,h,e] \otimes \mathcal{R}$ ($e \in [1, n]$, $\bar{e} \in [n+1, 2n]$) is included in the graph only when $\mathcal{R}$ already lists the element $e - 1$. Such a graph with complete vertex ordering generates a single instance of each diagram topology; the corresponding computational cost of the sum of all diagrams is $\mathcal{O}(n^2 2^n)$, exhibited by the actual number of algebraic operations in Fig. 3 (CoS-2, all vertices ordered). There we also plot the computational cost of summing the nonsymmetrised skeleton series in terms of $G$ and $W$, which is dramatically reduced compared to that of its symmetrised counterpart in Fig. 2.

We find that, in the case of the $SU(N)$ Hubbard model, the speed-up of the fully nonsymmetrised summation (CoS-2, all vertices ordered) relative to the partially nonsymmetrised sum (CoS-2, ordered interaction ends) fails to compensate the substantial increase of the corresponding Monte Carlo variance. We therefore use the latter variant for our calculation of the equation of state. However, the fully nonsymmetrised summation is inherently better compatible with algorithmic integration over imaginary time[53] or Matsubara frequency[54], as well as deterministic numeric integration, such as the tensor train approach[55], where the absence of symmetrisation should help reduce the entanglement between the vertices.

### Equation of state of the 2D SU(N) Hubbard model

The recent study by Pasqualetti et al.[38] has revealed a perfect agreement between the DQMC calculations and experimental measurements of the EoS—the average particle number $\langle n \rangle$ per lattice site as a function of the chemical potential $\mu$—of the 2D $SU(N)$ Hubbard model at temperatures down to $T/t = 0.3$ at $U/t = 2.3$ and coupling up to $U/t = 10.4$ at $T/t = 1.35$ for $N = 6$. As a benchmark, we obtain the $\langle n \rangle(\mu)$ curve at $T/t = 0.3$, $U/t = 2.3$, plotted in Fig. 4, and find it to be in perfect agreement with that computed and measured in ref. 38 (The EoS reported in ref. 38 is averaged over small intervals of the chemical potential due to the experimental resolution, but we have verified the agreement within combined error bars for the prime numerical data, kindly provided by the authors.).

Ibarra-García-Padilla et al.[23] demonstrate that the sign problem in their DQMC simulations rapidly intensifies with lowering $T$ and increasing $U$ and $N$ at the considered densities, as long as the system remains compressible. To explore the more challenging regime, the

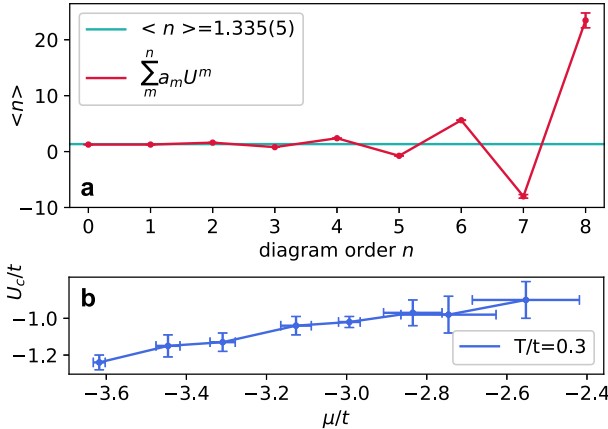

**Fig. 5 | Evaluating the series. a** Partial sum of the diagrammatic series for density $\langle n \rangle$ as a function of the truncation order $n$ for $N = 6$ and $T/t = 0.3$, $\mu/t = 0.575$, $U/t = 2.3$. The horizontal line is the result of a reconstruction of the value from the series, $\langle n \rangle = 1.335(5)$. **b** The location of the singularity $U_c$ responsible for the series divergence at $N = 6$ and $T/t = 0.3$ as a function of the chemical potential $\mu$ (corresponding to $\langle n \rangle(\mu) \sim 1$–2.5 in this range and at $U = U_c$). The error bars are propagated from the statistical (standard deviation) errors of evaluating the series coefficients. Source data are provided as a Source Data file.

EoS was obtained by DiagMC at a lower temperature $T/t = 0.15$ (see Fig. 4), for which the $\langle n \rangle(\mu)$ curve is below that for $T/t = 0.3$ at $U/t = 2.3$, indicating that the entropy density $s(\mu)$ is an increasing function in this range of $\mu$ due to the Maxwell relation $\partial s / \partial \mu = \partial n / \partial T$. Following ref. 56, this also suggests that the system is in the metallic state in this regime. We further evaluate the series at larger values of $U$ up to $U = 12$. A plateau around $\langle n \rangle = 1$ can be seen to start emerging at $U = 8$, consistently with the development of a (pseudo-)gapped state at strong coupling. The plateau appears more pronounced at $U = 12$, but at these couplings the systematic error of resummation beyond the convergence radius (see Methods) becomes comparable to the propagated statistical error, and the combined error bar, i.e. a sum of the two errors, shown in Fig. 4 grows substantially. Nonetheless, the analytic structure of the series appears free from singularities near a positive real $U$, such as those in the $SU(2)$ Hubbard model at similar parameters[56,57]. There, the growth of the antiferromagnetic (AFM) correlation length beyond $\sim 10$ lattice sites was shown to be responsible for a near-critical behaviour of the diagrammatic expansions at these temperatures and $\langle n \rangle = 1$ already at $U/t \sim 3$ (see the Supplemental Material of ref. 56 for details). Also in contrast to the $SU(3)$ case, where an insulating AFM ground state at $\langle n \rangle = 1$ emerges for $U/t \gtrsim 5.5$[29] and strong AFM correlations (with a transformation upon heating) are observed up to $T/t \sim 0.5$ at $U/t = 8$[24], for $N = 6$ AFM correlations appear weak down to $T/t = 0.15$ at this coupling, as implied by the absence of the corresponding singularities or a pronounced plateau in $\langle n \rangle(\mu)$. This is not unexpected since for larger $N$ the Fermi surface at $\langle n \rangle = 1$ is farther from being nested.

## Discussion

The introduced approach represents a versatile platform for evaluating Feynman's diagrammatic series: It is naturally applicable to fermionic as well as bosonic systems, to expansions in bare coupling and renormalised or skeleton series[1], to expansions derived from the homotopic action[13], in and out of equilibrium with the extension by the Keldysh formalism[58,59], and may find use in other advanced approaches based on the diagrammatic theory[55,60–63]. Being intrinsically division-free, the technique is compatible with diagrammatic methods based on algorithmic integration over Matsubara frequency[54] or imaginary time[53], in which dynamic correlation functions are computed directly

without the need for numerical analytic continuation, and an efficient way of summing the diagrams would be crucial for accessing strongly correlated regimes. The vector formulation of the algorithm is a promising foundation for realising DiagMC on a quantum computer by mapping the polynomial-size graph to a quantum circuit, with the Quantum DiagMC offering an exponential speed-up over the classical counterpart. On a classical computer, the exponential scaling of the number of operations needed to evaluate all terms of a given order places[11] the CoS approach in the class of numerical methods with polynomial computational complexity. The rigid graph structure lends itself to efficient hardware acceleration and parallelisation, while the partial summation and subtraction at intermediate levels of the graph reduce the bit complexity, making the algorithm robust against rounding errors.

The example application to the EoS of the $2D$ $SU(6)$ Hubbard model provides controlled benchmarks for ongoing theoretical and experimental studies, aimed at accessing lower temperatures and novel quantum many-body states. As a byproduct of a diagrammatic calculation, the analytic structure of the series offers additional insights into the physics of the system. The results suggest a phase transition in the attractive $SU(N)$ Hubbard model at a coupling strength as low as $U_c/t \sim -1$ up to temperatures $T/t \lesssim 0.5$ (see Methods), and absence of evidence for strong AFM correlations in the repulsive case at the considered temperatures and interaction strengths, at which the $SU(2)$[57,64] and $SU(3)$[24,29] Hubbard models are already in the (quasi-)AFM state. In the $SU(2)$ case, the formulation in the thermodynamic limit enabled DiagMC to attain controlled accuracy in the regime where correlations are intrinsically long-range and are difficult to capture reliably by finite-size methods even in the absence of the sign problem[57,64]. Such regimes of the $SU(N)$ model is where the developed technique can prove particularly useful. The possibility of a direct calculation of entropy in the DiagMC approach[56] could be instrumental for thermometry in experiments with ultracold atoms that are currently testing the limits of state-of-the-art theoretical methods.

## Methods

The average particle number per lattice site $\langle n \rangle$ is expressed in our approach as an expansion in the powers of the Hubbard coupling $U$, $\langle n \rangle(T, \mu, U) = \sum_{m=0}^{\infty} a_m(T, \mu) U^m$, in the thermodynamic limit. In DiagMC $\mu$ is typically shifted to improve convergence[65]; a $U$-dependent shift, described, e.g., in ref. 44 is adopted here. The series coefficients $a_m$ are obtained by the Monte Carlo integration of all connected diagram topologies of order $m$—summed here by the CoS-2 technique with ordered interaction ends in $\mathcal{O}(m^3 3^m)$ operations—over the positions of the $m$ interaction lines in space-imaginary-time. Thus, $a_m$ are known numerically exactly with statistical error bars, while the only source of systematic error is the truncation of the series at order $n$. Although the series turns out to be divergent in all regimes of interest, being able to evaluate $a_m$ up to $n = 8$ with $< 5\%$ statistical error (and fractions of percent at lower orders) enables an accurate reconstruction of the result with controlled precision.

Figure 5a shows the partial sum for the density of the $2D$ $SU(6)$ Hubbard model at $U/t = 2.3$, $\mu/t = 0.575$ and $T/t = 0.3$, the lowest temperature in the study by Pasqualetti et al.[38], as a function of the truncation order $n$. The series is seen to wildly diverge, but its analytic structure is rather simple, with the dominating singularity at $U_c/t \approx -0.9(1)$, which allows us to reconstruct the result following the approach developed in ref. 44. Specifically, we employ the Dlog-Padé[66] technique, taking into account the statistical error bars of $a_m$ and verifying that the systematic error of the evaluation—detected by the variation of the answer with free parameters of Dlog-Padé—is negligible in this regime. The result for the series in Fig. 5 is $\langle n \rangle = 1.335(5)$. The singularity at a real $U$ is indicative of a phase transition exhibited by the attractive $SU(N)$ Hubbard model, which is likely to a superfluid state. When the series for the relevant susceptibility is considered, the

divergence at $U_c$ is an accurate tool for characterising the critical point[67]. Leaving the calculation of susceptibilities for a more focused study, we plot in Fig. 5b a crude estimate of $U_c$ from the divergence of density at $T/t = 0.3$.

## Data availability

Source data are provided with this paper. Other data that support the findings of this study are available from the author upon request.

## Code availability

The codes that support the findings of this study are available from the author upon request.

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

## Acknowledgements

The author is grateful to Kaden Hazzard for illuminating and stimulating discussions, to Eduardo Ibarra-García-Padilla, Sohail Dasgupta, Kaden Hazzard, and Richard Scalettar for sharing their DQMC data, and to Sohail Dasgupta, Simon Fölling, Kaden Hazzard, Eduardo Ibarra-García-Padilla, Giulio Pasqualetti, and Richard Scalettar for a fruitful exchange of results and ideas. This work was supported by EPSRC through Grant No. EP/X01245X/1. The calculations were performed using King's Computational Research, Engineering and Technology Environment (CREATE).

## Author contributions

E.K. developed the algorithm and codes, carried out the calculations, analysed the results and wrote the paper.

## Competing interests

The author declares no competing interests.
