## [Peer Review File · Nature Communications]

Combinatorial summation of Feynman diagramsREVIEWER COMMENTS

Reviewer #1 (Remarks to the Author):

Report on Combinatorial summation of Feynman diagrams: Equation of state of the 2D SU(N) Hubbard model, Evgeny Kozik.

The manuscript introduces a means of organizing the diagrammatic expansion that is based on a tree of graphs. Each branching represents a contraction of operators, resulting in a Greens function. A set of rules is set p to generate the correct factor and sign of the diagram. While the naive summation of graphs generates a set of terms that grow factorially with order, this approach costs $\sim N^4$.

Normally, diagrammatic methods deal with expansions i connected topologies. In concurrent sampling techniques, disconnected graphs must be subtracted. In determinant methods, this is achieved by introducing counter terms, which add computational complexity. In this work, it is proposed that such counter terms may instead be implemented by modifying the tree.

The method is tested for a SU(6) Hubbard model, where the equation of state is obtained. An expansion to order N=8 is provided for interaction strengths ranging from $U/t=2.3$ to 8, the latter being in the strongly correlated regime. The series is divergent due to a singularity probably stemming from a Cooper instability on the attractive side, but resummation gives an accurate estimate for the equation of state.

This work presents ideas that are original, and in my view also valuable. The introduction of determinant methods in the past had a major impact on the applicability of diagrammatic MC, and this work shows that you can improve upon that method in terms of computational complexity. The method is presented in a clear way. I do therefore recommend that this work is published.

Reviewer #2 (Remarks to the Author):

Summary

The manuscript under review introduces an innovative and technically sophisticated approach for the efficient summation of Feynman diagrams, an important tool in quantum many-body perturbation theory. The authors have developed a framework that is applied to the Diagrammatic Monte Carlo (DiagMC) approach, specifically focusing on the SU(N)-symmetric Hubbard model. This framework represents a significant advancement in the field, addressing a complex theoretical challenge that has long been a hindrance in the study of many-body systems.

Major Contributions

1. **Innovative Framework for Feynman Diagrams:** The manuscript introduces a framework for the efficient summation of Feynman diagrams across a range of many-body systems. The methodological depth and thoroughness in the presentation of this framework are particularly impressive, considering the historical challenges associated with Feynman diagrams in quantum many-body physics.
2. **Methodological Advancement in Quantum Computing:** The exploration of potential applications of this framework in quantum computing is a forward-thinking and significant aspect of the paper. The possibility of achieving exponential speed-up in calculations over

classical methods presents a potential paradigm shift in computational techniques within this field.

3. New Insights into the SU(N) Hubbard Model: The application of this technique to the SU(N) Hubbard model not only demonstrates its utility but also provides new insights into the model, particularly in complex regimes such as lower temperatures and higher coupling strengths. This is a significant contribution to understanding the behavior of such systems, shedding light on phenomena like phase transitions and the absence of strong antiferromagnetic correlations for $N = 6$.

Key Areas for Revision

1. Enhancing Accessibility for a Multidisciplinary Audience: Currently, the manuscript is dense and highly specialized, potentially limiting its appeal to the broader, multidisciplinary audience of Nature Communications. To enhance accessibility, the authors should consider revising the manuscript to simplify some of the more complex technical discussions. A more general and accessible overview of the research's significance is suggested, particularly in the introduction, which currently starts with the specialized field of Diagrammatic Monte Carlo. A gentler introduction that builds up to this complex topic would be more suitable for a diverse readership.

2. Broadening Context and Implications: It is recommended that the authors expand the discussion on the broader implications of this research, particularly in the context of various fields within many-body physics. Highlighting how this advancement could influence other areas of study will underscore its multidisciplinary relevance and appeal.

Conclusion

The manuscript marks a significant contribution to quantum many-body physics. However, its technical density and specialized focus may not align well with the multidisciplinary scope of Nature Communications. A major revision is strongly recommended to enhance its accessibility.

The scientific core of the paper is undeniably robust. However, the manuscript requires a much more broad and approachable introduction, alongside clearer explanations of complex concepts and an expanded discussion on its broader implications in many-body physics. This is crucial for making the manuscript suitable for the diverse readership of Nature Communications.

The manuscript's fit for a multidisciplinary journal hinges on the authors' ability to effectively communicate their findings to a broader audience. These major revisions are essential for the paper's potential acceptance in Nature Communications.

The manuscript introduces a new way to efficiently compute series of connected Feynman diagrams, and a first application to the SU(6) Hubbard model, for which experimental data recently became available. Compared to existing diagrammatic Monte Carlo algorithms, the presented ‘‘Combinatorial summation’’ (CoS) approach brings a significant change of paradigm, and offers clear advantages in a significant number of important situations. Compared to the approach of Ref. [12] (Chen and Haule, Nat. Commun. 2019), the CoS approach is superior in terms of computational complexity, since the number of operations grows only exponentially (as opposed to factorially) with diagram order. Compared to the connected determinant (CDet) approach of Ref. [10] (Rossi, PRL 2017), the CoS approach is more flexible, since it can easily handle renormalized or skeleton series, and works in different representations (*e.g.* momentum instead of position representation); moreover, the number of operations for CoS is smaller than for CDet for certain models such as the SU(N) Hubbard model with $N > 2$. The presented computations, for the SU(6) Hubbard model in a strongly correlated low-temperature regime, demonstrate that the new approach goes beyond current state-of-the-art, for interesting models of direct relevance to current experiments.

In my opinion this makes the manuscript suitable for publication Nature Communications, provided the following points are addressed. Many of these points are recommendations to clarify certain aspects of the method. I understand that fully clarifying all these points is not possible within the space limits of the main text, and I suggest adding a Supplementary Information.

Main points

- (1) *‘‘It is easy to see that the number of floating point operations in this graph is only exponential, $\mathcal{O}(n^3)4^n$ ’’*
This doesn’t seem obvious to me. Since this is one of the major insights of this work, a derivation should be presented.
- (2) *‘‘The challenge of restricting the series to irreducible diagrams in both channels is met here by supplementing the nodes in the CoS-2 graph with a record \mathcal{R} that keeps track of connectivity when a propagator or interaction line is cut’’*
This is not really clear. Isn’t it rather ‘‘connectivity when a pair of propagator or interaction line is cut’’ ? This is an important feature of the method, and the procedure should be described explicitly.
- (3) *‘‘we proceed to obtain the $\langle n \rangle(\mu)$ curve at $T/t = 0.3$, $U/t = 2.3$, plotted in Fig. 4, and find it to be in perfect agreement’’* followed by the footnote: *‘‘The EoS reported in Ref. [38] is averaged over small intervals of the chemical potential due to the experimental resolution, but we have verified the agreement within combined error bars for the prime numerical data, kindly provided by the authors.’’*
It would be very nice to provide a plot (maybe in the Supplementary Information) showing this agreement with the experimental and numerical data from Ref. [38].
- (4) About the statement
‘‘a faint shoulder around $\langle n \rangle = 1$ is seen to emerge. This is consistent with the development of a (pseudo-)gapped state.’’
(and the corresponding statement at the very end of the introductory Section):
I am not sure to understand the logic behind this interpretation of the (indeed very faint) shoulder. In the Mott-insulating regime, for the $\langle n \rangle$ vs. μ curve, one expects a *plateau around $\langle n \rangle = 1$* . I understand that the plateau would only develop for significantly larger values of U . As a first stage towards developing the plateau, I would expect *two* shoulders, at densities slightly below and above unity. It would be interesting to investigate this by adding a few more data points around $\langle n \rangle = 1$.
- (5) Looking at the DQMC data in Fig. 1f of Ref. [23] for $U/t = 12$ (and T/t between 1 and 0.625), one sees a clear plateau developing around unit density, which means that there should be a significant change between $U/t = 8$ and 12. It would be interesting if the author could add data at larger U to investigate this. I understand that increasing U makes the computation harder, but if necessary this could be investigated at larger T , since the aforementioned data of Ref. [23] at $U/t = 12$ feature the incipient plateau at T/t as high as 1.
- (6) I wonder whether there is a simple physical explanation for the fact that there is almost no sign of the Mott plateau at $U/t = 8$: Is this expected given the relatively large value $N = 6$?
- (7) It would be very nice to extend to higher U the comparison with the experimental and numerical data of Ref. [38]. This would be facilitated by the fact that the corresponding T/t values are larger (they are roughly proportional to U , according to Fig. 2e of Ref. [38]). Even a comparison for a limited set of values of μ would be a useful benchmark.

- (8) Is there a good reason to include the factor 2 in the factor $(2n)^3$ in the expression for the number of operations for CDet, $(2n)^3 2^n + 3^n$, in Fig. 2a ? If not, it would be appropriate to remove it, which would anyways not change the main conclusions.
- (9) *“Nonetheless, the analytic structure of the series appears free from singularities near a positive real U , such as those in the $SU(2)$ Hubbard model at similar parameters [55]. There, the growth of the antiferromagnetic (AFM) correlation length beyond ~ 10 lattice sites was shown to be responsible for a near-critical behaviour of the diagrammatic expansions at these temperatures and $\langle n \rangle = 1$ already at $U/t \sim 3$. Also (...) for $N = 6$ AFM correlations appear weak down to $T/t = 0.15$ at this coupling.”*
- (a) I do not find any discussion in Ref. [55] about “singularities near near a positive real U ” or “near-critical behaviour of the diagrammatic expansions”.
- (b) The absence of singularities near the positive real U axis is the only reason presented in the manuscript for concluding that “AFM correlations appear weak” in the $SU(6)$ case, a conclusion which is also repeated in the final paragraph of the manuscript (“absence of strong AFM correlations in the repulsive case”). In my opinion this is a very indirect evidence, and in order to make a reliable statement about AFM correlations, one needs to compute the correlation function. Therefore, this conclusion should either be presented in a less definitive way, making it clear that it is based on this single indirect evidence, or it should be supported by actual computations of the correlation function.
- (10) *“Thus, there is no fundamental difficulty to reduce the errors bars and access larger U values at the expense of a polynomially-longer calculation.”*
This sounds like the computational time increases polynomially as a function of U , which is not the case, as far as I know. This wording should be changed to avoid this confusion (e.g. just remove “and access larger U values”).
- (11) *“At large N , however, unbiased numerical methods are currently outperformed by experimental realisations of the system with ultracold alkaline-earth-like atoms in optical lattices (...) in accessing the regimes of low temperatures and strong correlations [37,38].”*
If I understand correctly, in Ref. [38] the experimental data agree with DQMC up to $U/t = 10.4$, and for $U/t = 33$ the experimental data also agree with unbiased computational methods (numerical linked cluster expansion and high-temperature series expansion). Therefore, I don’t think that Ref. [38] is an example of experiment outperforming unbiased numerics.

Other points:

- (12) In the directed graphs of Fig. 1, what is the meaning of the colors of the edges?
- (13) It would be helpful to describe in more detail the procedure for setting to zero disconnected diagrams: How is “cutting the corresponding path” implemented ?
- (14) If I understand correctly, according to the procedure described at the end of Sec. II.A, the following two rules for constructing the graph are implicit (presumably also in II.B):
- (i) In each cycle, the head appears first. For example, (487) is allowed, but (874) is forbidden.
- (ii) The order of appearance of the cycles in a cycle cover is such that the cycle heads appear in increasing order. For example, a cycle cover of the form (1...)(4...) is allowed, but (4...)(1...) is forbidden.
- This deserves to be stated explicitly.
- (15) About the phrase: “now the two nodes (...) are considered identical, and are merged, only if $\mathcal{R}_1 = \mathcal{R}_2$ ”
What do you mean exactly by “are considered identical, and are merged” ? Is “merged” a reference to the pruning step described in the next sentence (“a sub-optimal graph can always be pruned in the end”) ?
- (16) About the phrase: “In principle, the information in \mathcal{R} should be kept minimal to prevent spawning redundant nodes, but a sub-optimal graph can always be pruned in the end (...)”
This could be made more clear. “In principle, the information in \mathcal{R} should be kept minimal” sound a bit convoluted; I suppose you rather mean something like “It is desirable to keep minimal the information in \mathcal{R} (...)”.

- (17) About “we also store in \mathcal{R} a matrix of connections between the cycles, which does not need to be optimal”, I have two questions:
- The phrase sounds like the matrix does not need to be optimal, but maybe you rather mean that \mathcal{R} does not need to be optimal?
 - Do you really include this in \mathcal{R} , in the sense that this will be used to tell whether two nodes are considered identical? Or do you rather store this in another record, which will *not* be used to tell whether two nodes are considered identical?
- (18) “To this end, \mathcal{R} only needs to store the list of interactions that a visited element belongs to, and whether only one vertex of the interaction or both have been visited, i.e. 3 possibilities for each interaction.”
There are several aspects of this sentence which I find confusing:
- It is not very clear what you mean by “interactions that a visited element belongs to”. I suppose that you mean: interactions such that one or both of its vertices have been visited.
 - If I understand correctly, the “3 possibilities” are: zero, one, or both vertices of the interaction have already been visited. However, the case “zero” should not appear if this is about interactions such that at least one of its vertices have been visited. So maybe it would be more appropriate to rewrite the entire sentence, for example as follows:
“To this end, \mathcal{R} only needs to store, for each interaction, the number its vertices (0, 1 or 2) which have already been visited”.
- (19) “making sure that the systematic error of the evaluation (...) is negligible.”
If I understand correctly what is meant, a more clear formulation would be, for example,
- to replace “making sure” by “we find”
 - to add at the end of the sentence “in this regime” (since this is not the case any more for the larger U values discussed later in the manuscript).
- (20) “at a lower temperature $T/t = 0.15$ (...) for which the $\langle n \rangle(\mu)$ curve is below that for $T/t = 0.3$, indicating that the system is in the metallic state at $U/t = 2.3$ and in this range of μ [54].”
Apparently, the following criterion is implicitly used here: if $(\partial \langle n \rangle / \partial T)_\mu > 0$ then the system is in the metallic state. I wonder where this criterion comes from. I did not find it in Ref. [54].
Furthermore, the difference between the two curves is very small (although the two temperatures differ by a factor two), which suggests that they are already close to the zero-temperature limit, so I wonder whether one can really conclude anything from the fact that one curve is (slightly) above the other.
- (21) “Ibarra-García-Padilla et al. [23] demonstrate that the sign problem in DQMC rapidly intensifies with lowering T and increasing U and N at the considered densities, as long as the system remains compressible.”
It may be more appropriate to replace “DQMC” by something like “their DQMC simulations”.
Indeed, in the article F. Assaad, PRB 71, 075103 (2005), one can find the following statement:
“The simplifications which occur in the large- N limit, namely the suppression of quantum fluctuations have important consequences for auxiliary field quantum Monte Carlo (QMC) simulations. As a function of growing values of N the negative sign problem inherent to stochastic methods is reduced thus rendering simulations more and more tractable.”
I understand that this statement has not been checked numerically in that work by F. Assaad, since it is restricted to half filling, which is sign free for the Hubbard Stratonovich transformation used in that work (while the sign problem is always present for the Hubbard Stratonovich transformation used in Refs. [38] and [23]).
Anyhow, it is not clear to me whether the growth of the sign problem when increasing N observed in [23] is a general feature of DQMC or whether it is specific to the Hubbard Stratonovich transformation used in [23].

Minor issues and typos:

- (22) “The recent study by Pasqualetti et al. [38] has revealed a perfect agreement between the DQMC calculations and experimental measurements of the EoS of the 2D $SU(N)$ Hubbard model down to $T/t = 0.3$ and a coupling value up to $U/t = 2.3$ for $N = 6$.”
I would remove “down to” and “up to”, because Ref. [38] actually reports agreement between DQMC and experiments up to $U/t = 10.4$ (for T/U of about 0.12 to 0.14).

- (23) The definition of “*weight* \mathcal{C} ” seems to be different in two places of the manuscript. From the statement “*the current weight* \mathcal{C} *is multiplied by* $g_{ee'}$, *as well as by an additional* -1 *when the cycle is closed*”, it seems that the factor $(-1)^m$ is included in “*weight* \mathcal{C} ”. But this is not the case in the equations “*sign* $\mathcal{C} = (-1)^{2n+m}$ *and* *weight* $\mathcal{C} = (g_{\alpha_1\alpha_2} \dots g_{\alpha_{c_1}\alpha_1}) \dots (g_{\alpha_{c_{m-1}+1}\alpha_{c_{m-1}+2}} \dots g_{\alpha_{2n}\alpha_{c_{m-1}+1}})$ ” appearing right after eq. (2).
- (24) “*there will be exactly one such \mathcal{C}' that”
I suppose you rather mean “there will be exactly one \mathcal{C}' such that”.*
- (25) After “*computing the determinant in* $\mathcal{O}(n^4)$ *floating-point operations*”, maybe cite again Ref. [43].
- (26) The acronym “*CDet*” is not defined anywhere in the manuscript.
- (27) Typo: “*in arbitrarily order*”
- (28) Replace “*the element*” by “an element” in “*if each new cycle starts with the element that is paired by an interaction (...)*” ?
- (29) About the endnote [68], “*In DiagMC* μ *is typically shifted to improve convergence* [65]; *a* U -*dependent shift, described, e.g., in Ref. [44] is adopted here, which explains the different range of* μ .”
It may be more appropriate to refer to this endnote (removing “*which explains the different range of* μ ”) already after the first sentence of Section III, since it means that the expansion does not exactly correspond to the formula “ $\langle n \rangle(T, \mu, U) = \sum_{m=0}^{\infty} a_m(T, \mu) U^m$ ” given in that sentence.

Reviewer #1 (Remarks to the Author):

Report on Combinatorial summation of Feynman diagrams: Equation of state of the 2D SU(N) Hubbard model, Evgeny Kozik.

The manuscript introduces a means of organizing the diagrammatic expansion that is based on a tree of graphs. Each branching represents a contraction of operators, resulting in a Greens function. A set of rules is set p to generate the correct factor and sign of the diagram. While the naive summation of graphs generates a set of terms that grow factorially with order, this approach costs $\sim N^4$.

Normally, diagrammatic methods deal with expansions i connected topologies. In concurrent sampling techniques, disconnected graphs must be subtracted. In determinant methods, this is achieved by introducing counter terms, which add computational complexity. In this work, it is proposed that such counter terms may instead be implemented by modifying the tree.

The method is tested for a SU(6) Hubbard model, where the equation of state is obtained. An expansion to order N=8 is provided for interaction strengths ranging from $U/t=2.3$ to 8, the latter being in the strongly correlated regime. The series is divergent due to a singularity probably stemming from a Cooper instability on the attractive side, but resummation gives an accurate estimate for the equation of state.

This work presents ideas that are original, and in my view also valuable. The introduction of determinant methods in the past had a major impact on the applicability of diagrammatic MC, and this work shows that you can improve upon that method in terms of computational complexity. The method is presented in a clear way. I do therefore recommend that this work is published.

I am grateful to Reviewer #1 for their positive assessment of the presented results and for recommending the paper for publication. The reviewer does not have any further questions.

Reviewer #2 (Remarks to the Author):

Summary

The manuscript under review introduces an innovative and technically sophisticated approach for the efficient summation of Feynman diagrams, an important tool in quantum many-body perturbation theory. The authors have developed a framework that is applied to the Diagrammatic Monte Carlo (DiagMC) approach, specifically focusing on the SU(N)-symmetric Hubbard model. This framework represents a significant advancement in the field, addressing a complex theoretical challenge that has long been a hindrance in the study of many-body systems.

Major Contributions

1. **Innovative Framework for Feynman Diagrams:** The manuscript introduces a framework for the efficient summation of Feynman diagrams across a range of many-body systems. The methodological depth and thoroughness in the presentation of this framework are particularly impressive, considering the historical challenges associated with Feynman diagrams in quantum many-body physics.

2. **Methodological Advancement in Quantum Computing:** The exploration of potential applications of this framework in quantum computing is a forward-thinking and significant aspect of the paper. The possibility of achieving exponential speed-up in calculations over classical methods presents a potential paradigm shift in computational techniques within this field.

3. **New Insights into the SU(N) Hubbard Model:** The application of this technique to the SU(N) Hubbard model not only demonstrates its utility but also provides new insights into the model, particularly in complex regimes such as lower temperatures and higher coupling strengths. This is a significant contribution to understanding the behavior of such systems, shedding light on phenomena like phase transitions and the absence of strong antiferromagnetic correlations for $N = 6$.

I am grateful to Reviewer #2 for their detailed overview of the different contributions of the paper and their favourable evaluation, as well as for the constructive and helpful suggestions, which are addressed below.

Key Areas for Revision

1. **Enhancing Accessibility for a Multidisciplinary Audience:** Currently, the manuscript is dense and highly specialized, potentially limiting its appeal to the broader, multidisciplinary audience of Nature Communications. To enhance accessibility, the authors should consider revising the manuscript to simplify some of the more complex technical discussions. A more general and accessible overview of the research's significance is suggested, particularly in the introduction, which currently starts with the specialized field of Diagrammatic Monte Carlo. A gentler introduction that builds up to this complex topic would be more suitable for a diverse readership.

I have extended the introduction to include a more general overview of the use of Feynman diagrams in a broader context across different fields of physics. The emphasis on Diagrammatic Monte Carlo (DiagMC) is now reduced and the introduction builds up to the subject more gradually.

Furthermore, addressing the specific comments of Reviewer #3, many of which are concerned with the accessibility of the technical arguments, should additionally help make the text more accessible for a diverse readership.

2. **Broadening Context and Implications:** It is recommended that the authors expand the discussion on the broader implications of this research, particularly in the context of various fields within many-body physics. Highlighting how this advancement could influence other areas of study will underscore its multidisciplinary relevance and appeal.

The discussion section does open with, in my view, a very broad overview of the immediate applications of the technique in general terms, without being limited to a particular field of many-body physics: “*The introduced approach represents a versatile platform for evaluating Feynman’s diagrammatic series: It is naturally applicable to fermionic as well as bosonic systems, to expansions in bare coupling and renormalised or skeleton series [5], to expansions derived from the homotopic action [13], in and out of equilibrium with the extension by the Keldysh formalism [59, 60], and may find use in other advanced approaches based on the diagrammatic theory [54, 61–64].*”

With the extended introduction, where the other areas of physics where Feynman diagrams are used are now explicitly listed, this should now clearly imply that the method could make an impact in those areas as well. (I would like to refrain from making more speculative statements.)

Conclusion

The manuscript marks a significant contribution to quantum many-body physics. However, its technical density and specialized focus may not align well with the multidisciplinary scope of Nature Communications. A major revision is strongly recommended to enhance its accessibility.

The scientific core of the paper is undeniably robust. However, the manuscript requires a much more broad and approachable introduction, alongside clearer explanations of complex concepts and an expanded discussion on its broader implications in many-body physics. This is crucial for making the manuscript suitable for the diverse readership of Nature Communications.

The manuscript's fit for a multidisciplinary journal hinges on the authors' ability to effectively communicate their findings to a broader audience. These major revisions are essential for the paper’s potential acceptance in Nature Communications.

I hope that the referee finds the revised manuscript suitable for publication in Nature Communications.

Reviewer #3 (Remarks to the Author):

The manuscript introduces a new way to efficiently compute series of connected Feynman diagrams, and a first application to the SU(6) Hubbard model, for which experimental data recently became available. Compared to existing diagrammatic Monte Carlo algorithms, the presented “Combinatorial summation” (CoS) approach brings a significant change of paradigm, and offers clear advantages in a significant number of important situations. Compared to the approach of Ref. [12] (Chen and Haule, Nat. Commun. 2019), the CoS approach is superior in terms of computational complexity, since the number of operations grows only exponentially (as opposed to factorially) with diagram order. Compared to the connected determinant (CDet) approach of Ref. [10] (Rossi, PRL 2017), the CoS approach is more flexible, since it can easily handle renormalized or skeleton series, and works in

different representations (e.g. momentum instead of position representation); moreover, the number of operations for CoS is smaller than for CDet for certain models such as the SU(N) Hubbard model with $N > 2$. The presented computations, for the SU(6) Hubbard model in a strongly correlated low-temperature regime, demonstrate that the new approach goes beyond current state-of-the-art, for interesting models of direct relevance to current experiments.

In my opinion this makes the manuscript suitable for publication Nature Communications, provided the following points are addressed. Many of these points are recommendations to clarify certain aspects of the method. I understand that fully clarifying all these points is not possible within the space limits of the main text, and I suggest adding a Supplementary Information.

I'm indebted to Reviewer #3 for their extremely careful reading of the manuscript, for recommending the paper for publication in Nature Communications and for their specific and detailed suggestions, which very much helped me improve the manuscript.

(1) "It is easy to see that the number of floating point operations in this graph is only exponential, $O(n^3)4^n$ "

This doesn't seem obvious to me. Since this is one of the major insights of this work, a derivation should be presented.

An explicit derivation of the complexity scaling has now been added: "*Indeed, at a given level l the number of nodes, i.e. unique labels $[l, h, e] \otimes R$, is a product of the number of values of h , e and R , given by $O(l)$, $O(2n - l)$ and $O((2n)!/l!(2n - l)!)$ respectively. Each node is linked to $O(n)$ nodes at the level $l + 1$, and thus the total number of edges is obtained by summing $n l(2n - l)(2n)!/l!(2n - l)!$ over all levels l , resulting in the exponential scaling.*" The computational complexity of all the modifications of the algorithm is derived identically and should now be obvious. Since the derivation is really that simple, I think it is useful to keep it in the main text (and there is no need for a Supplementary Information).

(2) "The challenge of restricting the series to irreducible diagrams in both channels is met here by supplementing the nodes in the CoS-2 graph with a record R that keeps track of connectivity when a propagator or interaction line is cut"

This is not really clear. Isn't it rather "connectivity when a pair of propagator or interaction line is cut" ? This is an important feature of the method, and the procedure should be described explicitly.

The referee is right, this wasn't explained very well. The specific rules for filtering the diagrams in the G-W expansion are now stated explicitly: "The challenge of restricting the series to irreducible diagrams in both channels is met here by supplementing the nodes in the CoS-2 graph with a record R that keeps track of connectivity—as in the CoS-1 approach of Sec. II B—when any two propagators in a cycle or two interaction lines between different cycles are cut."

(3) “we proceed to obtain the $\langle n \rangle(\mu)$ curve at $T/t = 0.3$, $U/t = 2.3$, plotted in Fig. 4, and find it to be in perfect agreement” followed by the footnote: “The EoS reported in Ref. [38] is averaged over small intervals of the chemical potential due to the experimental resolution, but we have verified the agreement within combined error bars for the prime numerical data, kindly provided by the authors.”

It would be very nice to provide a plot (maybe in the Supplementary Information) showing this agreement with the experimental and numerical data from Ref. [38].

I intended to do this originally, but it turns out there are at least two reasons that make showing the comparison with the experiment/numerics of Ref. [38] less useful than it might seem. Firstly, the experimental and numerical data for $\langle n \rangle(\mu)$ reported in Ref. [38] is averaged over small intervals of $\Delta\mu$. A direct comparison would require averaging the EoS obtained here too, which largely hides possible discrepancies and would be inappropriate and potentially misleading for the purpose of benchmarking. A more scientifically valid approach is to compare directly with the unprocessed numerical (DQMC) data of Ref. [38]. However, the DQMC results there carry a substantial error bar, which was sufficient for comparing with the experiment, but is less impressive if shown alongside the present DiagMC results. So the second reason is that, if the prime DQMC data of Ref. [38] were plotted against the results of this paper, this would imply that the DQMC method is considerably less accurate than it actually is, which would be misleading (and unfair to the authors of Ref. [38]).

To perform a truly definitive comparison, the authors of Ref. [38] kindly offered to carry out additional DQMC calculations reducing the error bars for several data points and were able to reach a comparable accuracy at $U/t=2.3$ to that reported here. The DiagMC and DQMC data agreed perfectly with the reduced DQMC error bars as well. It wouldn't be possible to report the better DQMC calculations here due to the authorship of the data, but a separate joint work focused on cross-benchmarking state-of-the-art methods for the SU(N) Hubbard model is planned.

(4) About the statement

“a faint shoulder around $\langle n \rangle = 1$ is seen to emerge. This is consistent with the development of a (pseudo-)gapped state.”

(and the corresponding statement at the very end of the introductory Section):

I am not sure to understand the logic behind this interpretation of the (indeed very faint) shoulder. In the Mott-insulating regime, for the $\langle n \rangle$ vs. μ curve, one expects a plateau around $\langle n \rangle = 1$. I understand that the plateau would only develop for significantly larger values of U . As a first stage towards developing the plateau, I would expect two shoulders, at densities slightly below and above unity.

The use of “shoulder” here was probably ambiguous in this context; this sentence is now replaced by: “A plateau around $\langle n \rangle = 1$ can be seen to start emerging at $U = 8$, consistently with the development of a (pseudo-)gapped state at strong coupling.”

It would be interesting to investigate this by adding a few more data points around $\langle n \rangle = 1$.

I have added more points around $\langle n \rangle = 1$ and reduced the error bars in this regime. This would have been a prohibitively lengthy calculation, but it turned out that the efficiency of the CoS algorithm could be further reduced in the case of the Hubbard interaction. Specifically, the diagrammatic series generated by the original algorithm is automatically symmetrised over exchanges of the end points on each interaction line. This is redundant for the Hubbard model because the interaction is strictly local in both space and time. Therefore, the direction of the interactions can be ordered – which reduces the computational complexity of the graph from $O(n^3 4^n)$ to $O(n^3 3^n)$ – without any increase in the Monte Carlo variance.

The symmetrisation of the sum of all order- n diagrams over $n!$ permutations of the interaction lines can also be easily suppressed, reducing the computational cost further to $O(n^2 2^n)$. This, however, is not very useful when the integration over vertex positions is done by Monte Carlo because the increase of the Monte Carlo variance due to the lack of the symmetrisation is not compensated by the speed-up. I have added a very brief Sec. IIE explaining this.

With the added data points and smaller error bars, the development of the plateau can now be seen more clearly.

(5) Looking at the DQMC data in Fig. 1f of Ref. [23] for $U/t = 12$ (and T/t between 1 and 0.625), one sees a clear plateau developing around unit density, which means that there should be a significant change between $U/t = 8$ and 12. It would be interesting if the author could add data at larger U to investigate this. I understand that increasing U makes the computation harder, but if necessary this could be investigated at larger T , since the aforementioned data of Ref. [23] at $U/t = 12$ feature the incipient plateau at T/t as high as 1.

I have now evaluated the updated $T/t=0.15$ series at $U/t=12$ (see Fig. 5). Although the error bars are notably increased compared to the $U/t=8$ case, the plateau at $\langle n \rangle = 1$, as well as the difference between the gradients of $\langle n \rangle(\mu)$ for $\langle n \rangle < 1$ and $\langle n \rangle > 1$, is more pronounced now.

The problem with comparing to the DQMC data of Ref. [23] is that there the results were obtained only on a 6×6 lattice. When the insulating state is driven by extending AFM correlations, finite-size systems are known to be strongly biased towards exhibiting the insulating behaviour. A spectacular illustration in the case of the $SU(2)$ Hubbard model at $\langle n \rangle = 1$ is offered by Fig. 2 of Supplemental Material of [Phys. Rev. B 91, 125109 (2015)], where the metal-insulator crossover is characterised by Blankenbecler-Scalapino-Sugar quantum Monte Carlo (QMC) calculations on lattices up to 16×16 sites: For $T/t=0.1$ and $U/t=2$ the system on a 8×8 lattice is an insulator with a pretty large gap. However, when the system size reaches beyond 12×12 (presumably substantially larger than the AFM correlation length) the metallic character is recovered.

Thus, cross-benchmarking DiagMC and DQMC calculations in a correlated regime is a major endeavour, which is planned for a separate work in collaboration with DQMC experts.

(6) I wonder whether there is a simple physical explanation for the fact that there is almost no sign of the Mott plateau at $U/t = 8$: Is this expected given the relatively large value $N = 6$?

I think it is expected. Note that in the SU(2) case at $\langle n \rangle = 1$, the ground state is an antiferromagnetic insulator for any $U > 0$ due to the perfect nesting of the Fermi surface. It was shown in Ref. [29] (of the revised manuscript) that already for $N=3$, which breaks nesting at $\langle n \rangle = 1$, an AFM insulating ground state appears only for $U > 5.5t$. It is thus natural to expect that for $N=6$ the critical U will be pushed to an even higher value due to an even more circular Fermi surface.

To make this point clearer, I have added (after "...for $N=6$ AFM correlations appear weak down to $T/t = 0.15$ at this coupling."): "*This is not unexpected since for larger N the Fermi surface at $\langle n \rangle = 1$ is farther from being nested.*"

(7) It would be very nice to extend to higher U the comparison with the experimental and numerical data of Ref. [38]. This would be facilitated by the fact that the corresponding T/t values are larger (they are roughly proportional to U , according to Fig. 2e of Ref. [38]). Even a comparison for a limited set of values of μ would be a useful benchmark.

I have added the $U/t=12$ data at $T/t=0.15$, which is perhaps more relevant for demonstrating the power and limitations of the method. The calculations at $T/t > 1$ at large U , as in Ref. [38], are simple and a lot less instructive in this context but would probably be more useful in a separate study with a greater emphasis on the underlying physics (where the AFM correlations could be studied explicitly by computing the spin-spin correlation function).

(8) Is there a good reason to include the factor 2 in the factor $(2n)^3$ in the expression for the number of operations for CDet, $(2n)^3 2^n + 3^n$, in Fig. 2a ? If not, it would be appropriate to remove it, which would anyways not change the main conclusions.

It seems more appropriate to keep the factor of 2 in the computational cost $(2n)^3$ of computing a determinant in CDet because the actual number of operations (rather than just the scaling) is plotted for the CoS approach there, which has to be computed for a $2n \times 2n$ matrix at the diagram order n .

Admittedly, a direct comparison is difficult because the practical number of operations for determinant calculation depends on the implementation and hardware optimisation; hence $(2n)^3 2^n + 3^n$ is referred to as "theoretical scaling" according to Ref. [10].

(9) "Nonetheless, the analytic structure of the series appears free from singularities near a positive real U , such as those in the SU(2) Hubbard model at similar parameters [55]. There, the growth of the antiferromagnetic (AFM) correlation length beyond ~ 10 lattice sites was shown to be responsible for a near-critical behaviour of the diagrammatic expansions at these temperatures and $\langle n \rangle = 1$ already at $U/t \sim 3$. Also (. . .) for $N = 6$ AFM correlations appear weak down to $T/t = 0.15$ at this coupling."

(a) I do not find any discussion in Ref. [55] about "singularities near a positive real U " or "near-critical behaviour of the diagrammatic expansions".

I'm grateful to the referee for pointing out this inaccuracy – although the mentioned observation was indeed made when analysing the data for Ref. [55] ([58] in the modified text) it actually wasn't discussed there until a follow-up paper (PRL 126, 105701 (2021), Ref.

[57]), where a detailed discussion can be found in the Supplemental Material: “*Nonetheless, since the coefficients a_m have error bars, we are often unable to resolve the singularity with a purely real U_s from a pair of complex-conjugate singularities close to the real axis. Such a situation occurs at half-filling when the antiferromagnetic (AFM) correlation length extends beyond ~ 10 lattice sites [9].*”, where Ref. [9] is Ref. [55] of the first version of the manuscript.

This is addressed in the revised manuscript by the addition of “(see the Supplemental Material of Ref. [57] for details)” at the end of the sentence starting with “*There, the growth of the antiferromagnetic (AFM) correlation length...*”

(b) The absence of singularities near the positive real U axis is the only reason presented in the manuscript for concluding that “AFM correlations appear weak” in the SU(6) case, a conclusion which is also repeated in the final paragraph of the manuscript (“absence of strong AFM correlations in the repulsive case”).

In my opinion this is a very indirect evidence, and in order to make a reliable statement about AFM correlations, one needs to compute the correlation function. Therefore, this conclusion should either be presented in a less definitive way, making it clear that it is based on this single indirect evidence, or it should be supported by actual computations of the correlation function.

In view of the mechanism discussed above, it is difficult to imagine a scenario in which a long correlation length would not manifest itself in the singularity structure as well as in an apparent (quasi-) gapped behaviour (a plateau as a function of the chemical potential) of the equation of state. Nonetheless, I must agree with the referee that this is rather indirect evidence and, formally, the conclusion could be softened.

In the revised text, the statement “absence of strong AFM correlations” is replaced with “absence of evidence for strong AFM correlations”.

(10) “Thus, there is no fundamental difficulty to reduce the errors bars and access larger U values at the expense of a polynomially-longer calculation.”

This sounds like the computational time increases polynomially as a function of U , which is not the case, as far as I know. This wording should be changed to avoid this confusion (e.g. just remove “and access larger U values”).

Indeed, I can see – thanks to the referee – that this is confusing. The statement was supposed to mean that there are no singularities near the positive real U (as discussed above) and hence by reducing the error bars one should be able to access larger U values without a major difficulty for the series re-summation. However, this is a somewhat technical and relatively minor point, so I’ve removed this phrase in the revised text.

(11) “At large N , however, unbiased numerical methods are currently outperformed by experimental realisations of the system with ultracold alkaline-earth-like atoms in optical lattices (...) in accessing the regimes of low temperatures and strong correlations [37,38].”

If I understand correctly, in Ref. [38] the experimental data agree with DQMC up to $U/t = 10.4$, and for $U/t = 33$ the experimental data also agree with unbiased computational methods (numerical linked cluster expansion and high-temperature series expansion). Therefore, I don't think that Ref. [38] is an example of experiment outperforming unbiased numerics.

My understanding was that the experiment in Ref. [38] had not been pushed beyond state-of-the-art numerics, but had the capacity in principle. This appears to follow, in particular, from the use of the experiment as a benchmark for numerics and not the other way around as is usually the case: (e.g., in the abstract) "*Our measurements are used as a benchmark for state-of-the-art numerical methods including determinantal quantum Monte Carlo and numerical linked cluster expansion.*" Hence, it felt appropriate to give the experiment credit in this context.

(12) In the directed graphs of Fig. 1, what is the meaning of the colors of the edges?

The colours are merely used to mark the edges that emerge from the same node. In the end this is probably not so important because the example graphs are not very dense.

(13) It would be helpful to describe in more detail the procedure for setting to zero disconnected diagrams: How is "cutting the corresponding path" implemented ?

I'm not entirely sure what additional detail could be useful in the discussion of removing the disconnected diagrams. It consists of two parts, one outlining the general idea:

"Since there is a one-to-one correspondence between a particular path in the graph and the diagram it generates, the task of omitting the disconnected diagrams from the determinant can be formulated as that of identifying the corresponding paths and eliminating them selectively. Preserving all other paths is in principle accomplished by duplicating certain nodes along the unwanted paths and re-routing the paths to be kept through the copies, as in the example in Fig. 1d."

and the other going through a detailed example of how this principle is implemented in practice, in the paragraph starting with: "*A disconnected diagram is produced when not all of its cycles (fermionic loops) end up linked by the interaction lines...*"

Perhaps addressing the following comments, which are concerned with some details of this discussion, addresses the referee's concern here.

(14) If I understand correctly, according to the procedure described at the end of Sec. II.A, the following two rules for constructing the graph are implicit (presumably also in II.B):

(i) In each cycle, the head appears first. For example, (487) is allowed, but (874) is forbidden.

Yes, this is correct.

(ii) The order of appearance of the cycles in a cycle cover is such that the cycle heads appear in increasing order. For example, a cycle cover of the form (1 . . .)(4 . . .) is allowed, but (4 . . .)(1 . . .) is forbidden.

This is also correct.

This deserves to be stated explicitly.

Indeed, this follows from the rules for constructing the graph in the last paragraph of Sec. IIA but wasn't explicitly declared. This is now stated in the revised manuscript: "*However, it was demonstrated in Ref. [43] that all terms with repeated elements will cancel out due to*

the sign structure, provided the lowest element in each cycle within C , called the cycle head h , starts the cycle and is present in C only once.", and also "*so that the cycle heads are picked in ascending order.*" at the end of the sentence starting with "This is straightforward to ensure..."

(15) About the phrase: "now the two nodes (. . .) are considered identical, and are merged, only if $R_1 = R_2$ "

What do you mean exactly by "are considered identical, and are merged" ? Is "merged" a reference to the pruning step described in the next sentence ("a sub-optimal graph can always be pruned in the end") ?

The use of "merged" here could indeed be confused with the pruning procedure. The words "and are merged" are now removed and the sentence revised to make this unambiguous: "*If what constitutes R is identified, the right graph can be constructed from the start by the algorithm of Sec. II A with the modification that the two nodes $[l_1, h_1, e_1] \otimes R_1$ and $[l_2, h_2, e_2] \otimes R_2$ are considered one and the same if $R_1 = R_2$ in addition to $l_1 = l_2, h_1 = h_2, e_1 = e_2$.*"

(16) About the phrase: "In principle, the information in R should be kept minimal to prevent spawning redundant nodes, but a sub-optimal graph can always be pruned in the end (...)"

This could be made more clear. "In principle, the information in R should be kept minimal" sound a bit convoluted; I suppose you rather mean something like "It is desirable to keep minimal the information in R (...)".

That's exactly right. The sentence is now revised: "It is desirable that the information in R is kept minimal to prevent spawning redundant nodes, but..."

(17) About "we also store in R a matrix of connections between the cycles, which does not need to be optimal", I have two questions:

(a) The phrase sounds like the matrix does not need to be optimal, but maybe you rather mean that R does not need to be optimal?

Indeed, it's the record R that need not be optimal, which already follows from "and prune the final graph". In the revised text, "which does not need to be optimal" is removed to avoid the confusion.

(b) Do you really include this in R, in the sense that this will be used to tell whether two nodes are considered identical ? Or do you rather store this in another record, which will not be used to tell whether two nodes are considered identical ?

It is the former: the additional information about connections between the cycles is included in R in the sense that it will be used to tell whether two nodes are considered one and the same, which should now be clear.

(18) "To this end, R only needs to store the list of interactions that a visited element belongs to, and whether only one vertex of the interaction or both have been visited, i.e. 3 possibilities for each interaction."

There are several aspects of this sentence which I find confusing:

- It is not very clear what you mean by "interactions that a visited element belongs to". I suppose that you mean: interactions such that one or both of its vertices have been visited.
- If I understand correctly, the "3 possibilities" are: zero, one, or both vertices of the interaction have already been visited. However, the case "zero" should not appear if this is about interactions such that at least one of its vertices have been visited. So maybe it would be more appropriate to rewrite the entire sentence, for example as follows:

"To this end, R only needs to store, for each interaction, the number its vertices (0, 1 or 2) which have already been visited".

The referee is right and I'm grateful for this suggestion, which is a much simpler way of putting it. The sentence is now modified almost exactly as suggested by the referee: "*To this end, R only needs to store, for each interaction line, the number of its vertices—zero, one or two—that have already been visited.*"

(19) "making sure that the systematic error of the evaluation (. . .) is negligible."

If I understand correctly what is meant, a more clear formulation would be, for example,

- to replace "making sure" by "we find"

That is correct. To clarify this, "making sure" is now replaced with "verifying".

- to add at the end of the sentence "in this regime" (since this is not the case any more for the larger U values discussed later in the manuscript).

Implemented.

(20) “at a lower temperature $T/t = 0.15$ (. . .) for which the $\langle n \rangle(\mu)$ curve is below that for $T/t = 0.3$, indicating that the system is in the metallic state at $U/t = 2.3$ and in this range of μ [54].”

Apparently, the following criterion is implicitly used here: if $(\partial \langle n \rangle / \partial T)_\mu > 0$ then the system is in the metallic state. I wonder where this criterion comes from. I did not find it in Ref. [54].

This indeed comes from (i) the observation that $(\partial s / \partial \mu)_T = (\partial \langle n \rangle / \partial T)_\mu > 0$ and (ii) the fact that, as argued in Ref. [54], the crossover to the non-Fermi-liquid regime can be associated with the inflection point in $s(\mu)$, which generally takes place already when $(\partial s / \partial \mu) < 0$.

Furthermore, the difference between the two curves is very small (although the two temperatures differ by a factor two), which suggests that they are already close to the zero-temperature limit, so I wonder whether one can really conclude anything from the fact that one curve is (slightly) above the other.

For instance, one conclusion, which could be useful, e.g., for experiments is that $(\partial s / \partial \mu)_T > 0$ in this regime. I’ve tried to make both of these points clearer in the revised text: “*To explore the more challenging regime, the EoS was obtained by DiagMC at a lower temperature $T/t = 0.15$ (see Figure 5), for which the $\langle n \rangle(\mu)$ curve is below that for $T/t = 0.3$ at $U/t = 2.3$, indicating that the entropy density $s(\mu)$ is an increasing function in this range of μ due to the Maxwell relation $\partial s / \partial \mu = \partial n / \partial T$. Following Ref. [57], this also suggests that the system is in the metallic state in this regime.*”

(21) “Ibarra-García-Padilla et al. [23] demonstrate that the sign problem in DQMC rapidly intensifies with lowering T and increasing U and N at the considered densities, as long as the system remains compressible.”

It may be more appropriate to replace “DQMC” by something like “their DQMC simulations”. Indeed, in the article F. Assaad, PRB 71, 075103 (2005), one can find the following statement:

“The simplifications which occur in the large- N limit, namely the suppression of quantum fluctuations have important consequences for auxiliary field quantum Monte Carlo (QMC) simulations. As a function of growing values of N the negative sign problem inherent to stochastic methods is reduced thus rendering simulations more and more tractable.”

I understand that this statement has not been checked numerically in that work by F. Assaad, since it is restricted to half filling, which is sign free for the Hubbard Stratonovich transformation used in that work (while the sign problem is always present for the Hubbard Stratonovich transformation used in Refs. [38] and [23]). Anyhow, it is not clear to me whether the growth of the sign problem when increasing N observed in [23] is a general feature of DQMC or whether it is specific to the Hubbard Stratonovich transformation used in [23].

The referee is absolutely right: the structure of the sign problem in DQMC does depend on the kind of the Hubbard-Stratonovich transformation used and it appears not impossible that another variant of the DQMC method could describe this regime better. Following the

suggestion of the referee, “DQMC” has been replaced by “their DQMC simulations” in the revised text.

(22) “The recent study by Pasqualetti et al. [38] has revealed a perfect agreement between the DQMC calculations and experimental measurements of the EoS of the 2D SU(N) Hubbard model down to $T/t = 0.3$ and a coupling value up to $U/t=2.3$ for $N =6$.”

I would remove “down to” and “up to”, because Ref. [38] actually reports agreement between DQMC and experiments up to $U/t = 10.4$ (for T/U of about 0.12 to 0.14).

The sentence has been corrected accordingly: “*The recent study by Pasqualetti et al. [38] has revealed a perfect agreement between the DQMC calculations and experimental measurements of the EoS of the 2D SU(N) Hubbard model at temperatures down to $T/t =0.3$ at $U/t = 2.3$ and coupling up to $U/t = 10.4$ at $T/t = 1.35$ for $N = 6$.*”

(23) The definition of “*weight C*” seems to be different in two places of the manuscript. From the statement “the current *weight C* is multiplied by $g_{ee'}$, as well as by an additional -1 when the cycle is closed”, it seems that the factor $(-1)^m$ is included in “*weight C*”. But this is not the case in the equations “*sign C* = $(-1)^{2n+m}$ and *weight C* = $(g_{a_1 a_2} \dots g_{a_{c-1} a_c}) \dots (g_{a_{cm-1+1} a_{cm-1+2}} \dots g_{a_{2nacm-1+1}})$ ” appearing right after eq. (2).

This is now corrected by replacing “the current *weight C*” by “the current *sign C* · *weight C*”.

(24) “there will be exactly one such C' that”

I suppose you rather mean “there will be exactly one C' such that”.

Yes, this has now been corrected.

(25) After “computing the determinant in $O(n^4)$ floating-point operations”, maybe cite again Ref. [43].

Implemented.

(26) The acronym “CDet” is not defined anywhere in the manuscript.

The first mention of “CDet” is now replaced by “*connected determinant diagrammatic Monte Carlo (CDet)*”.

(27) Typo: “in arbitrarily order”

Corrected.

(28) Replace “the element” by “an element” in “if each new cycle starts with the element that is paired by an interaction (...)” ?

Corrected.

(29) About the endnote [68], “In DiagMC μ is typically shifted to improve convergence [65]; a U-dependent shift, described, e.g., in Ref. [44] is adopted here, which explains the different range of μ .”

It may be more appropriate to refer to this endnote (removing “which explains the different range of μ ”) already after the first sentence of Section III, since it means that the expansion does not exactly correspond to the formula “ $\langle n \rangle(T, \mu, U) = \sum a_m(T, \mu) U^m$ ” given in that sentence.

This is a good point; the endnote has been moved to the first sentence of Section III.

REVIEWERS' COMMENTS

Reviewer #2 (Remarks to the Author):

Thank you for the revised manuscript. I appreciate the author's efforts to improve the accessibility of the introduction for a multidisciplinary audience. The expanded overview of Feynman diagrams in a broader context is a welcome addition.

Overall, the manuscript presents innovative and valuable research in quantum many-body systems. The combinatorial summation technique introduced here represents a significant advancement with potential wide-ranging applications in physics.

In conclusion, I recommend this manuscript for publication in Nature Communications. The author's efforts to improve accessibility, combined with the work's scientific merit, make it a valuable contribution that should interest a wide audience.

Reviewer #2 (Remarks on code availability):

The link provided is not working, possibly due to a permission setting issue.

I believe that the revised manuscript is significantly improved, as a result of the substantial extra work carried out by the author, including the addition of new data. There are just two of my points which the author did not properly address, in my opinion. Assuming that he will do so, I can recommend publication.

- Regarding point (8) from my first report:

After a careful analysis, I conclude that the formula $(2n)^3 2^n$ used by the author is really too naive.

First, the matrix of size $2n$ by $2n$ matrix has a special structure: It is composed of two matrices (one for each spin state) of size n (or n plus a constant in presence of external legs). Therefore, for a naive computation of all principal minors of both of these matrices, the number of operations is bounded by $2n^3 2^n$, instead of $(2n)^3 2^n$.

Moreover, the fast principal minor algorithm allows one to reduce this number of operations to $O(2^n)$, as was demonstrated in [PRB 105, 125104 (2022)]. The precise formula for the number of operations is given in that same reference, $\sum_{l=0}^{n-2} 2^l \times 2 \times (n-l-1)^2$. The author should use the latter formula, since this is the state of the art implementation of the CDet method.

- Regarding point (9b) from my first report:

I insist that it would be appropriate to clarify on which evidence the statement “AFM correlations appear weak” is based on (namely the absence of plateau in the $\langle n \rangle$ vs. μ curve and the absence of singularities near the real positive axis). Otherwise, some readers could be misled to believe that the spin correlation function was computed.

Note: I have not reviewed the code since the provided URL [<https://github.com/Kozik-Group/CoS>] does not exist.

Remaining suggestions of Reviewer #3:

- Regarding point (8) from my first report:

After a careful analysis, I conclude that the formula $(2n)^{32^n}$ used by the author is really too naive.

First, the matrix of size $2n$ by $2n$ matrix has a special structure: It is composed of two matrices (one for each spin state) of size n (or n plus a constant in presence of external legs). Therefore, for a naive computation of all principal minors of both of these matrices, the number of operations is bounded by $2n^{32^n}$, instead of $(2n)^{32^n}$.

Moreover, the fast principal minor algorithm allows one to reduce this number of operations to $O(2^n)$, as was demonstrated in [PRB 105, 125104 (2022)]. The precise formula for the number of operations is given in that same reference, $\sum 2^l 2(n-l-1)^2$. The author should use the latter formula, since this is the state of the art implementation of the CDet method.

Response: The manuscript has been revised accordingly: the formula suggested by the referee for the number of operations to compute the principal minors is now used for the reference line in Fig. 2a. This formula is also stated explicitly in the caption together with a reference to [PRB 105, 125104 (2022)] and the fact that this is the state-of-the-art implementation of the CDet algorithm (as opposed to the original general formulation given before).

- Regarding point (9b) from my first report:

I insist that it would be appropriate to clarify on which evidence the statement “AFM correlations appear weak” is based on (namely the absence of plateau in the $\langle n \rangle$ vs. μ curve and the absence of singularities near the real positive axis). Otherwise, some readers could be misled to believe that the spin correlation function was computed.

Response: I have now added after “AFM correlations appear weak down to $T/t=0.15$ at this coupling,” the following clarification: “as implied by the absence of the corresponding singularities or a pronounced plateau in $\langle n \rangle (\mu)$ ”.